# Effect of Feeding Cold-Pressed Sunflower Cake on Ruminal Fermentation, Lipid Metabolism and Bacterial Community in Dairy Cows

**DOI:** 10.3390/ani9100755

**Published:** 2019-10-01

**Authors:** Izaro Zubiria, Aser Garcia-Rodriguez, Raquel Atxaerandio, Roberto Ruiz, Hanen Benhissi, Nerea Mandaluniz, Jose Luis Lavín, Leticia Abecia, Idoia Goiri

**Affiliations:** 1NEIKER-Granja Modelo de Arkaute, Apdo. 46., 01080 Vitoria-Gasteiz, Spain; izaro23@hotmail.com (I.Z.); ratxaerandio@neiker.eus (R.A.); rruiz@neiker.eus (R.R.); hanening@gmail.com (H.B.); nmandaluniz@neiker.eus (N.M.); 2CICbioGUNE. Parque Científico Tecnológico de Bizkaia, Ed. 801A. Derio, 48160 Bizkaia, Spain; jllavin@cicbiogune.es (J.L.L.); labecia@cicbiogune.es (L.A.)

**Keywords:** biohydrogenation, sunflower cake, microbiota

## Abstract

**Simple Summary:**

The use of cold-pressed sunflower cake, a by-product of small-scale biodiesel manufacturing, as a substitute for prilled palm fat in dairy cows’ diet, can reduce the extent of unsaturated fatty acid biohydrogenation This favors an accumulation of vaccenic acid in the rumen concomitant with a greater daily duodenal microbial N flow and all without impairing ruminal fermentation, microbial diversity or abundance of dominant populations. In the present study, only changes in relative abundances of less-representative genera were induced.

**Abstract:**

Cold-pressed sunflower cake (CPSC), by-product of oil-manufacturing, has high crude fat and linoleic acid concentrations, being a promising supplement to modulate rumen fatty acid (FA) profile. This trial studied CPSC effects on ruminal fermentation, biohydrogenation and the bacterial community in dairy cows. Ten cows were used in a crossover design with two experimental diets and fed during two 63-day periods. The cows were group fed forage ad libitum and the concentrate individually. The concentrates, control and CPSC, were isoenergetic, isoproteic and isofat. The ruminal samples collected at the end of each experimental period were analyzed for short-chain fatty acid, FA and DNA sequencing. CPSC decreased butyrate molar proportion (4%, *p* = 0.005). CPSC decreased C16:0 (28%, *p* < 0.001) and increased C18:0 (14%, *p* < 0.001) and total monounsaturated FA, especially C18:1 trans-11 (13%, *p* = 0.023). The total purine derivative excretion tended to be greater (5%, *p* = 0.05) with CPSC, resulting in a 6% greater daily microbial N flow. CPSC did not affect the diversity indices but increased the relative abundances of *Treponema* and *Coprococcus*, and decreased *Enterococcus*, *Ruminococcus* and *Succinivibrio*. In conclusion, the changes in ruminal fermentation and the FA profile were not associated with changes in microbial diversity or abundance of dominant populations, however, they might be associated with less abundant genera.

## 1. Introduction

In recent decades, a growing interest in the search for nutritional strategies that modulate the fatty acid (FA) composition of milk fat has arisen. The goal is to reduce the saturated hypercholesterolemic FAs and increase the concentration of naturally occurring bioactive unsaturated FAs (UFA), such as rumenic acid (C18:2 cis-9 trans-11; RA), isomer of conjugated linoleic acid (CLA), and its metabolic precursor vaccenic acid (C18:1 trans-11; VA) [1].

The milk FA composition is mostly determined by the complex interactions between dietary factors and rumen metabolism [2]. The manipulation of the biohydrogenation process of dietary UFA offers the opportunity to modify the lipid composition of milk by changing the availability of FA for mammary uptake [3]. The lipid supplementation of ruminant diets has been reported to be the most straightforward way to modulate the rumen biohydrogenation process and thus the lipid profile of milk [4]. The enrichment of ruminant diets with polyunsaturated FA (PUFA) rich fats or oils is susceptible to decrease the level of atherogenic saturated FA (SFA), such as palmitic acid (C16:0) in rumen [5] and milk fat [6]. Besides, it can reduce the extent of UFA biohydrogenation, favoring a high accumulation of VA in the rumen [7,8] and promoting flow from the rumen into the bloodstream. This C18:1 isomer acts as a precursor for the endogenous synthesis of RA in mammary gland through Δ9-desaturation [9].

The adoption of feeding systems based on the utilization of PUFA-rich plant by-products in animal diets instead of fats used in human food would represent a good strategy to improve the nutritional value of milk fat, while reducing food-feed competition and livestock feeding costs. Cold-pressed sunflower cake (CPSC) is a cheap by-product of oil-manufacturing which can constitute an important human-inedible fat resource for livestock production [10,11]. This oily cake can be obtained on-farm after simple mechanical extraction of the oil. In addition to its high protein content, CPSC has a higher crude fat content than those of conventional solvent and expeller meals (up to 230 g/kg compared with 30 and 100 g/kg, respectively; [12]) and contains significant amounts of linoleic acid [13]. This makes it an attractive lipid supplement to increase the energetic density of livestock diets and coping with the need for modulating FA metabolism in the rumen. Interestingly, the few available data on the use of CPSC to modulate biohydrogenation process have reported promising results attributable to its linoleic acid (C18:2 cis-9 cis-12) concentration. In this respect, a recent in vitro study [10] reported that, included into a high-concentrate diet, CPSC was able to interfere with FA metabolism in the rumen, decreasing the ruminal concentration of SFA and increasing the accumulation of bioactive VA. The same study suggested that these changes in the rumen FA profile might be associated with impaired ruminal fermentation due to a change in the growth and activity of rumen microorganisms by CPSC-PUFA. However, to the best of the authors knowledge, no studies have examined if these effects on the biohydrogenation process, microbial protein synthesis, ruminal fermentation and bacterial community would be extended to in vivo conditions.

Therefore, this study was conducted to analyze the effect of using CPSC as an alternative lipid supplement in dairy cows’ diet on the biohydrogenation of dietary unsaturated FA, ruminal fermentation and bacterial community composition.

## 2. Materials and Methods

All experimental procedures were performed in accordance with the European Union Directive (2010/63/EU) and Spanish Royal Decree (RD 53/2013) for the protection of animals used for experimental and other scientific purposes, and approved by the ethics committee (NEIKER-OEBA-2015–011).

### 2.1. Animals and Experimental Design

All cows were kept at the experimental research farm of Fraisoro Agricultural School (Zizurkil, Northern Spain) in loose housing conditions. A total of 4 lactating Holstein and six Brown Swiss dairy cows were paired based on breed, parity, days in milk, and milk yield during a two-week period. The average days of the milk, body weight (BW), and milk yield of the cows before the beginning of the experiment were (mean ± SD): 106 ± 37 d, 600.6 ± 63.8 kg and 26.3 ± 6.3 kg/d, respectively. All cows were fed the control diet (Table 1) during the covariate period and were randomly assigned (within pair) to the control (CTR) or experimental diets (CPSC) in a cross-over design. Each experimental period lasted for 63 d. The first 48 d were allowed for adaptation to the diets and the following 15 d for measurements. The concentrates were formulated to provide similar amounts of energy, crude protein (CP) and fat (Table 1). The cows within a pair received the same amount of concentrate (mean 5 kg/d) in individual troughs offered at three different times per day (07:00, 15:00 and 21:00), but they had free access to a basal roughage mixed ration (Table 1). The cows were fed on mixed roughage ad libitum and CTR concentrate for an interval of 30 d between each experimental period, to avoid a carryover effect.

The cows were milked with an automated milking system (AMS, DeLaval, 2004, Tumba, Sweden) twice per day. All cows had free access to the AMS and were granted milking permission after 11 h from previous milking. In general, for any particular cow, when the time elapsed since last milking was more than 12 h during the day, that cow would be fetched and forced to visit the AMS.

### 2.2. Sampling and Measurements

The basal roughage and concentrates were sampled every two weeks. In week eight of each experimental period during four consecutive days, spot urine samples (approximately 300 mL each) were collected from each cow at 12:00 and 00:00 (d 1 of each sampling period), 9:00 and 21:00 (d 2), 06:00 and 18:00 (d 3) and 03:00 and 15:00 (d 4). The urine samples were collected by massaging the vulva. The urine was acidified (pH < 3) using 2M H_2_SO_4_. In week nine of each experimental period, the samples of ruminal content were taken four times over two consecutive days for short chain volatile fatty acid (SCFA) analysis, FA profile determination and DNA extraction for bacterial community studies. The sampling began at 00:00 and 12:00 on d 1, and 06:00 and 18:00 on d 2. The samples of the ruminal content were collected from each dairy cow using a stomach tube (18 mm diameter and 160 mm long) connected to a mechanical pumping unit (Vacuubrand ME 2SI, Wertheim, Germany). The ruminal content was filtered through four layers of sterile gauzes to separate the solid and liquid fractions. A 100 mL pool was made with 25 mL of the liquid fraction of each ruminal extraction for the FA profile study. Besides, another 100 mL of liquid fraction of each ruminal extraction were placed into a container for bacterial composition analysis. Finally, 15 mL filtered of each ruminal extraction were separated in individual tubes for SCFA analysis. All samples, included those from the rumen solid fraction, were immediately stored frozen at −20 °C ± 5 °C until analysis. The animal BW was determined on the first and last day of each experimental period using an automated weighing scale.

### 2.3. Sample Handling and Laboratory Procedures

#### 2.3.1. Feed

The roughage and concentrate were dried in a forced-air oven (48h, 60 °C), ground through a 1-mm sieve, and composited by period and animal. The dry matter (method 934.01) and N (method 984.13) contents were determined following [14]. The neutral detergent fibre was determined by the method of Van Soest et al. [15] with use of an alpha amylase, but without sodium sulphite, and was expressed free of ash. The acid detergent fibre, expressed exclusive of residual ash, was determined by the method of Robertson and Van Soest [16]. The fat content was determined without hydrolysis by the automated soxhlet method (Soxtec System HT 1043 Extraction Unit, Madrid, Spain) using hexane for 6 h as a solvent. The starch content was measured by polarimetry [17]. Fatty methyl esters (FAME) of lipid in 200 mg of freeze-dried concentrate samples were prepared in duplicate following a 1-step extraction-transesterification procedure, then methyl esters were separated and quantified by gas chromatography as explained below for samples of ruminal digesta. The peak identification was based on retention time comparisons with commercially available standard FAME mixtures (NU-Check Prep, Elysian, MN; Sigma-Aldrich, Madrid, Spain).

#### 2.3.2. Purine Derivatives Determination

The composited urine samples by animal and period were centrifuged, diluted (1:10), filtered (0.22 µm Millipore filter) and analyzed for purine derivatives (PD) by high performance liquid chromatography (HPLC), using a Shimadzu HPLC system equipped with a UV detector (205 nm) and two C18 reversed-phase columns (250 mm × 4.60 mm) connected in a series with the mobile phase NH4H2PO4-acetonitrile (80:20) gradient at a variable flow rate between 1.0–1.4 mL/min according to the method of Reynal Broderick [18], and using a 0.03 *M* KH2PO4 buffer solution and using allopurinol as an internal standard for the quantification. The peaks were identified by comparing their retention times with those of known standards as described in [18].

#### 2.3.3. Short Chain Fatty Acid Determination

The analysis of SCFA (acetic, propionic, butyric, isobutyric, valeric and isovaleric) of liquid fraction of the samples of ruminal digesta was performed by gas chromatography using a flame ionization detector. A volume of 4 mL of rumen liquor mixed with one mL of a solution of 20 g/L of ortophosphoric acid and 4 g/L of crotonic acid as an internal standard, in 0.5 N HCl, was centrifuged (15,000× *g* for 15 min at 4 °C) to separate the liquid phase from the feed residuals. After, the liquid phase was microfiltered (premium syringe filter regenerated cellulose, 0.45 μm 4 mm, Agilent Technologies, Madrid, Spain), and one mL of liquid phase was directly injected in the HPLC apparatus (Perkin-Elmer Inc., Boston, MA) using a semi capillary column (300 mm × 7.8 mm; 9-μm particle size; TR-FFAP, Supelco, Barcelona, Spain) and kept at 250 °C in the injector with a helium flow rate of 13 mL/min. The analyses were carried out applying an isocratic elution (flux 0.6 mL/min) with a 0.008 N H_2_SO_4_ solution as mobile phase. The injection loop was 20 μL. The individual SCFA were identified using a standard solution of 4.50 mg/mL of acetic acid, 5.76 mg/mL of propionic acid, 7.02 mg/mL of butyric acid and isobutyric acid, 8.28 mg/mL of valeric acid and isovaleric acid in 0.1 N H_2_SO_4_ (338826, 402907, B103500, 58360, 75054, 129542, respectively; Sigma-Aldrich, Madrid, Spain) The quantification was done using an external calibration curve based on the standards described above.

#### 2.3.4. Ruminal Fatty Acid Profile Determination

The ruminal FA profile determinations were performed as described in Toral et al. [19]. Lipid in 200 mg of freeze-dried samples of ruminal content was extracted with a mixture of hexane and isopropanol (3:2, vol/vol) [20] and converted to fatty acid methyl ester by sequential base-acid catalysed transesterification [21]. The total FAME profile was determined by gas chromatography using a flame ionization detector. The total FAME profile was determined using a temperature gradient program and then, isothermal conditions at 170 °C to further resolve 18:1 isomers [20]. The peaks were identified based on retention time comparisons with the same FAME mixtures used for the analysis of feeds. Other commercially available standards (from Nu-Chek Prep.; Sigma-Aldrich; and Larodan, Solna, Sweden), cross referencing with chromatograms reported in the literature e.g., [20,21], and a comparison with reference samples for which the FA composition, was determined based on gas chromatography analysis of FAME and gas chromatography-mass spectrometry analysis of corresponding 4,4-dimethyloxazoline derivatives [21].

#### 2.3.5. DNA Extraction and Illumina Library Generation

Prior to performing the DNA extraction, the samples of ruminal digesta were fractionated in liquid and solid fractions and lyophilized (Alpha 1–4 LD Plus, Christ, Osterode am Harz, Germany). The DNA extraction from a mixture based on dry mater weight of liquid and solid fractions of the samples of ruminal content was performed using the commercial Power Soil DNA Isolation kit (Mo Bio Laboratories Inc, Carlsbad, CA, USA) following the manufacturer’s instructions. The integrity of the DNA was checked on 0.8% (wt/vol) agarose gels and the yield and purity of extracted DNA were determined using a NanoDrop spectrophotometer (NanoDrop® ND-1000, Thermo Fisher Scientific) checking the 260/280 nm ratio. The purity of DNA was considered acceptable with ratios between 1.8–2.0.

The extracted DNA was subjected to paired-end Illumina sequencing of the V4 hypervariable region of the 16S rRNA genes [22]. The libraries were generated by means of Nextera kit. The 300bp paired-end sequencing reactions were performed on a MiSeq platform (Illumina, San Diego, CA, USA).

The forward and reverse reads were merged using FLASH-1.2.11 [23] with 10bp minimum overlap and allowing outie orientation, after which quality filtering was performed using the open-source software package QIIME (v.1.9.0): Quantitative Insights into Microbial Ecology software package [24]. The subsequent analysis, in addition to picking operational taxonomical units (OTU), assigning taxonomy, inferring phylogeny and creating OTU tables, were also performed by QIIME software [24]. The sequences were clustered as operational taxonomic units (OTUs) of 97% similarity using UCLUST [25]. The OTUs were checked for chimeras using the RDP gold data-base and assigned taxonomy using the Greengenes database [26]. The alpha and beta diversity metrics were calculated using the QIIME pipeline.

The sequence data have been deposited in the European Nucleotide Archive database under the accession number PRJEB33443.

### 2.4. Calculations and Statistical Analysis

The urinary excretion of allantoin and uric acid (PD) was used to estimate duodenal microbial N flow [27]. The total excretion of creatinine, urea, allantoin, allopurinol and uric acid for each daily interval was computed as the product of the urine volume obtained and metabolite concentration. One mean daily creatinine excretion rate (29.0 mg/kg of BW per d) was computed with the data from all the cows in the trial. The total PD excretion was the sum of allantoin and uric acid excreted in urine. The endogenous PD excretion (mmol/d) was estimated from the BW of individual cows as: 0.385 mmol/BW^0.75^ per d. The total absorption of microbial purines and ruminal synthesis of microbial N were calculated as described by Valadares et al. [28].

The data were analysed using the MIXED procedure of SAS [29]. Each cow was considered as the experimental unit (n = 10). The statistical model included the fixed effects of the treatment, breed, period and sequence and the random effect of the cow within the pair. The least squares means for treatments were reported. The treatment means were separated using a Tukey test except for the rumen FA profile where a Bonferroni adjustment was used. Significant effects were declared at *p* ˂ 0.05.

The significances between experimental groups on bacterial community were tested by analysis of dissimilarity (ADONIS) with 999 permutations. The significant fold changes of the OTU’s were performed in DESeq2 [30] based on a false discovery rate (FDR).

To investigate the correlations between the ruminal SCFAs or FAs and bacterial taxa, a regularized canonical correlation analysis (rCCA) was performed using the package mixOmics (v6.6.2) [31] in R (v3.5.1) [32]. To perform the rCCA analysis, the correlation values between the relative abundances of bacterial taxa (at genus level) and each ruminal FA or SCFA were computed to calculate a similarity matrix. A clustered image map and bipartite network were inferred using a similarity matrix obtained from the rCCA. In the bipartite network, these values were projected onto the space spanned by the first components retained in the analysis. Three relevant components were obtained setting a threshold to R = 0.45 for FA and R = 0.55 for SCFA.

## 3. Results

### 3.1. Rumen Fatty Cid Composition

Table 2 shows the SFA composition of ruminal contents when the cows were fed either the CTR or the CPSC diets. Although the total SFA did not differ between treatments (76.9 versus 76.2 g/100 g FA, *p* = 0.421), feeding CPSC induced marked changes in the rumen SFA profile, which were mainly characterized by a significant decrease in the proportion of C14:0 (2.29 versus 1.27 g/100 g FA, *p* < 0.001) and C16:0 (21.9 versus 15.7 g/100 g FA, *p* < 0.001), and an increase in the accumulation of C18:0 (44.0 versus 51.3 g/100 g FA, *p* < 0.001).

The responses of ruminal monounsaturated FA (MUFA) and PUFA to dietary treatments are reported in Table 3. The ruminal concentration of C18:1 cis-9 did not differ among treatments (4.25 versus 3.99 g/100 g FA, *p* = 0.512). Feeding CPSC increased C18:1 trans-11 concentration (4.83 versus 5.56 g/100 g FA, *p* < 0.001) and C18:1 trans-10 level (0.76 versus 0.94 g/100 g FA, *p* = 0.001) without altering the C18:1 trans-10/trans-11 ratio (0.16 versus 0.17, *P* = 0.165). The total cis MUFA remained statistically unmodified (6.2 versus 6.3 g/100 g FA, *p* = 0.794) but total trans MUFA rose substantially in response to CPSC feeding (10.7 versus 12.6 g/100 g FA, *p* < 0.001), resulting in a higher total MUFA when the cows were fed CPSC (16.9 versus 18.9 g/100 g FA, *p* = 0.009).

As observed in Table 3, the most abundant non-conjugated PUFA was linoleic acid (C18:2 cis-9 cis-12), but its ruminal concentration was not modified by dietary treatment (2.50 versus 2.39 g/100 g FA, *p* = 0.694). Contrarily, the ruminal concentration of other, less abundant non-conjugated dienes such as C18:2 cis-9 trans-12; C18:2 trans-11 cis-15 and C18:2 trans-11 trans-15, were higher with CPSC (*p* = 0.001; *p* = 0.006 and *p* < 0.001, respectively). Regarding conjugated C18:2 FA, both rumen digesta showed a similar amount of C18:2 cis-9 trans-11 CLA (0.474 versus 0.429 g/100 g FA, *p* = 0.589) and C18:2 trans-10 cis-12 CLA (0.069 versus 0.066 g/100 g FA, *p* = 0.858). However, a higher C18:2 trans-11 trans-13 CLA proportion was found in CPSC-digesta (0.171 versus 0.227 g/100 g FA, *p* = 0.009). The percentages of long-chain n-3 PUFA (0.663 versus 0.885 g/100 g FA, *p* = 0.05) and the n-6 PUFA (0.079 versus 0.175 g/100 g FA, *p* < 0.001) were increased when the cows were fed CPSC. Overall, feeding the CPSC diet modified the FA profile of rumen digesta toward a higher n-6:n-3 ratio (0.117 versus 0.208, *p* = 0.006).

### 3.2. Ruminal Fermentation and Microbial Protein Synthesis

The responses of the ruminal fermentation parameters and microbial protein synthesis to experimental treatments are presented in Table 4. Feeding CPSC did not affect molar proportions of acetate (66.5 versus 67.0 mmol/100 mmol, *p* = 0.108), propionate (16.9 versus 16.9 mmol/100 mmol, *p* = 0.940), isobutyrate (0.90 versus 0.88 mmol/100 mmol, *p* = 0.696), valerate (1.41 versus 1.43 mmol/100 mmol, *p* = 0.546) or isovalerate (1.06 versus 1.05 mmol/100 mmol, *p* = 0.835), but decreased the molar proportion of butyrate by 4 % (13.2 versus 12.7 mmol/100 mmol, *p* = 0.005). The acetate:propionate ratio was not affected by the dietary inclusion of CPSC (3.94 versus 3.97, *p* = 0.669). 

The daily excretion of allantoin (318 versus 335 mmol/d, *p* = 0.093) or uric acid (27 versus 29 mmol/d, *p* = 0.439) did not differ between treatments, but total PD tended to be greater with CPSC (345 versus 364 mmol/d, *p* = 0.05). Creatinine did not differ between treatments. The daily microbial N flow was 6 % greater when the cows were fed CPSC (257 versus 273 mmol/d, *p* = 0.045).

### 3.3. Ruminal Bacterial Community

Figure 1 represents the bacterial community composition in the rumen of cows when fed the two dietary treatments. The three most abundant phyla were Bacteroidetes (55.6%), Firmicutes (32.6%) and Proteobacteria (2.8%). The predominant family of Bacteroidetes was Prevotellaceae (predominant genus: Prevotella, (34.8%). Within Firmicutes, the dominant families in order of importance were Lachnospiraceae (11.8%), undefined families within the order of the Clostridiales (9.1%), Ruminococcaceae (7.5%) and Veillonellaceae (1.3%), whereas the Proteobacteria mainly consisted of undefined genera within the family Succinivibrionaceae (1%).

The experimental concentrate with CPSC did not influence bacterial species richness as expressed by different diversity indices, such as chao1 or shannon (Table 5). The beta diversity and the statistical test performed with ADONIS revealed no differences in bacterial community between experimental concentrates, but significant effects of breed and period were observed.

The OTUs with significant differences between treatments are shown in Figure 2. Due to the observed significant effect of period on bacterial community, only data from the first period were presented. The OTUs of the genera *Treponema* within the order Spirochaetales, *Cropococcus* within the order Clostridiales and some OTUs of the genera *Prevotella* within the order Bacteroidales were enriched in the ruminal digesta when the animals were fed the experimental concentrate with CPSC. On the other hand, in the ruminal digesta when cows were fed the control concentrate, an enrichment was observed in the OTUs of the genera Pediococcus, Lactobacillus, Weissella and *Enterococcus* within the order of Lactobacillales, *Bacillus*, *Paenibacillus* and *Rummeliibacillus* within the order of Bacillales, *Roseburia* and *Clostridium* within the order of Clostridiales, *Paludibacter* and some OTUs of the genera *Prevotella* within the order Bacteroidales, *Succinivibrio* within the order of Aeromonadales, *Burkholderia* and *Achromobacter* within the order of Burkholderiales, *Erwinia* and *Trabulsiella* within the order of Enterobacteriales and *Stenotrophomonas* within the order of Xanthomonadales.

The relationships between clusters of bacterial genera and rumen SCFA irrespective of experimental group are represented in Appendix A. This bipartite network was based on the regularized canonical correlations between relative bacterial abundances and relative concentrations of rumen SCFA. Strong significant correlations were observed, most of them related to branched-chain SCFA (BCSCFA). Genera *Prevotella* and undefined genera of the order Rickettsiales were positively correlated with BCSCFAs, whereas genera *Ruminococcus* and undefined genera of the order Clostridiales were negatively correlated. The undefined genera of the order Clostridiales were positively correlated with acetate proportions whereas genera *Acetobacter* and genera *Stenotrophomonas* were negatively correlated. The former two genera were found to be positively correlated with butyrate proportions. The propionate and valerate proportions presented weaker correlations.

The correlations between rumen FA and bacterial taxa were represented by a clustered image map (Figure 3) inferred from the rCCA analysis. In general, FA presented weaker correlations with bacterial taxa than SCFA. Genera *Shuttleworthia*, *Lachnospira, Treponema*, *Prevotella*, *Desulfovibrio*, undefined genera of the order Rickettsiales and family Veillonellaceae were positively correlated with some biohydrogenation intermediates (C18:1 trans-11, C18:1 trans-6-7-8-9-10-12-15, C18:1 cis-12, C18:2 trans-11 cis-15 and 13-oxo-C18:0), whereas these FA were negatively correlated with genera *Moryella*, *Mogibacterium*, *Sphaerochaeta*, *Anaeroplasma*, *RFN20*, undefined genera of the order Clostridiales and Bacteroidales and families Coriobacteriaceae, F16, Pirellulaceae, S24-7, Paraprevotellaceae, Christensenellaceae and RF16. Genera *Coprococcus* and *Pseudobutyrivibrio* were positively correlated with C18:2 trans-9 cis-12, C18:2 cis-9 trans-11 CLA and 10-oxo-C18:0, whereas undefined genera of the family Pirellulaceae were negatively correlated with these FA. Genera *Desulfovibrio*, *Acetobacter*, *Stenotrophomonas* and undefined genera within the families Veillonellaceae and Enterobacteriaceae were positively correlated with C18:1 cis-15, whereas genera *Sphaerochaeta* and *Anaeroplasma* were negatively correlated.

## 4. Discussion

The use of lipid sources (e.g., vegetable oils and oilseeds), that can be destined for human consumption and biodiesel production, to modify the nutritional quality of milk fat may imply competition between humans and animals for the same resources. The implementation of strategies that use local alternative lipid resources, such as CPSC, may contribute to reduce this competition. As these cakes can be produced at a farm level, they also promote low input milk production systems. The modulation of milk FA composition through the inclusion of CPSC in ruminant diet seems to be explained by the action of its PUFA on ruminal biohydrogenation of C18 FA [13]. However, the relationships between this action and potential consequences on ruminal fermentation and bacterial community composition remain uncertain. This study provides a description of the effects of using CPSC, a by-product of sunflower oil manufacturing, as an alternative lipid supplement in dairy cows’ diets on ruminal biohydrogenation of dietary FA, fermentation and bacterial communities.

### 4.1. Ruminal Biohydrogenation

The feeding of CPSC induced relevant changes in ruminal concentration of palmitic acid and stearic acid, the major SFA in ruminal fluid. The proportion of C16:0 in the rumen digesta of the experimental groups mimicked that of their diets. However, although C18:0 was offered in the diets at a low proportion (2.90% to 3.67% of total FA), it represented the major FA in the rumen digesta (44.0 to 51.30% of total FA), while the C18 UFAs, which represented the major FAs in experimental diets, were detected in low proportions in the rumen digesta. The high proportion of C18:0 and the concomitant lower proportion of C18 UFAs in rumen fluid indicate that a considerable amount of dietary C18 UFA was subjected to biohydrogenation, since C18:0 is the end product of biohydrogenation of these FAs [33]. The higher proportion of C18:0 found in CPSC-rumen digesta can be attributed to the fact that the CPSC diet supplied greater amounts of C18 UFAs, as compared with the CTR diet. Previous in vitro reports examining the effect of diets containing CPSC reported similar changes in the proportion of C16:0 and C18:0 in rumen digesta [10]. Although to date, no data is available regarding the FA profiles of milk from dairy cows fed CPSC, lower amounts of C16:0 and C18:0 have been reported in milk fat of sheep supplemented with CPSC [13].

In relation to C18:1 isomers, the response in the proportion of bioactive vaccenic acid (C18:1 trans-11) would support the potential inhibitory effect of CPSC on ruminal metabolism of UFA, which may improve the lipid profile of ruminant-derived products. The increase of this FA would specifically corroborate the hypothesis suggested by some authors that, due to its elevated PUFA content, CPSC can reduce the extent of the last step of C18 UFA biohydrogenation, increasing the ruminal outflow of C18:1 trans-11 to be deposited in ruminant products [10]. In fact, the measurements of milk FA composition ascertained that CPSC feeding significantly enhanced the VA concentration of dairy sheep milk fat [11,13]. This positive effect of CPSC on ruminal or milk concentration of VA agrees with the effects observed in other studies using sunflower oil [8,34]. The increase of the level of this FA in rumen digesta, and hence in milk fat, would point towards a nutritionally healthier product. The main ruminal species known to be involved in the final step of biohydrogenation (conversion of C18:1 trans-11 to 18:0) is *Butyrivibrio proteoclasticus* [35]. The present study was not able to identify changes at the species level, but although non-significant changes were observed between treatments in the abundance of *Butyrivibrio*, a higher abundance of those genera in cows fed the control concentrate was observed when the probability was not adjusted to FDR (Appendix A). Several other bacterial genera have been suggested to be involved in the conversion of C18:1 trans-11 to C18:0, including *Propionibacterium acnes, Selenomnomas ruminantium, Enterococcus faecium, Staphylococcus sp., Flavobacerium sp.* and *Streptococcus* [33]. However, among these bacterial genera, only the genus *Enterococcus* was negatively affected by CPSC in the present study. Regarding the bacterial taxa that positively correlated with VA (Figure 3), genera *Treponema* and *Prevotella* were significantly enriched with CPSC (Figure 2) and also genera *Lachnospira* was enriched with CPSC when the probability was not adjusted (Appendix A). This suggests some kind of relationship of bacterial species belonging to this taxa with changes induced by CPSC on biohydrogenation of C18 UFAs. In this sense, genera *Treponema* were previously reported to possess the ability to hydrogenate C18:2 to C18:1 trans-11 [36], and genera *Lachnospira* were reported to be positively correlated with rumen C18:1 trans-11 proportions [37], supporting this hypothesis.

Concerning the C18:1 trans-10, the increase detected in the ruminal concentration of this FA as a response to the dietary inclusion of CPSC was in agreement with recent in vitro findings [10]. Despite this increase, the C18:1 trans-10/ trans-11 ratio remained unmodified among treatments, indicating that no expressive trans-10 shift had occurred in the rumen biohydrogenation pathways [8]. The correlations of this FA with bacterial taxa were in line with those observed with C18:1 trans-11, suggesting that the same bacterial taxa probably were involved in the observed changes which affected in a similar way, the two biohydrogenation pathways. The observed ratio is in disagreement with the trans-10 shift reported in dairy cows supplemented with sunflower oil [34]. The reasons for this discrepancy between C18:1 trans-10/ C18:1 trans-11 ratio responses to supplementary sunflower lipids remain unclear, but it could be partially related to the form of lipid supplements. It is possible that, in contrast to free fats or oils, feeding seed coat protected lipids, such as CPSC, could have prevented to some extent the alteration of the rumen environment conditions and biohydrogenation pathways by dietary UFA, although differences between concentrates composition cannot be ruled out.

On the level of PUFA, the lack of a response of ruminal C18:2 cis-9 cis-12 to CPSC was consistent with the negligible changes observed in the proportion of this C18 UFA in the rumen of dairy cows receiving sunflower oil [8]. This result can be explained by the great extent of biohydrogenation that occurs with linoleic acid in the rumen (up to 95%; [38]).The increase in the ruminal C18:2 trans-11 cis-15 and C18:2 trans-11 trans-15 found with CPSC did not seem to be related to the changes in bacterial community composition, since none of the taxa less enriched in CPSC cows were correlated with these FA. This could probably be explained by a lower extent of linolenic acid biohydrogenation due to higher dietary UFA supply, considering that these non-conjugated C18:2 isomers are major metabolites of C18:3 cis-9 cis-12 cis-15 biohydrogenation [39]. Regarding conjugated dienes, the results of the current study have shown that the use of CPSC as an alternative lipid supplement in dairy cows’ diet failed to enhance the rumen concentration of C18:2 cis-9 trans-11 CLA, although genera *Coprococcus* that were enriched in CPSC cows was positively correlated with this FA. Nevertheless, the increased accumulation of ruminal C18:1 trans-11, the main precursor of C18:2 cis-9 trans-11 CLA synthesis in mammary tissue, would possibly increase the concentration of this health-promoting CLA isomer in milk fat [40]. In fact, a significant increase in the accumulation of C18:2 cis-9 trans-11 CLA has been detected in the milk fat of dairy sheep supplemented with CPSC [13]. In contrast to C18:2 cis-9 trans-11 CLA, the proportion of C18:2 trans-11 trans-13 CLA, an intermediate of linolenic acid biohydrogenation [1], was higher in CPSC rumen fluid. This result confirmed the ability of CPSC to reduce the extent of C18:3 cis-9 cis-12 cis-15 biohydrogenation and suggests a specific effect of this by-product on the microbiota involved in the cited process. In this sense, among the bacterial taxa described in literature playing a role in the main biohydrogenation pathways for linolenic acid [41], in the current study, genera *Ruminococcus* appeared less prevalent in CPSC diets, suggesting its role in the observed changes.

Regarding n3 FA, our results agree with previous studies that have demonstrated that supplementing with oils rich in C18:2n-6 and C18:3-n3 decrease the biohydrogenation of long-chain n-3 FA in the rumen [42].

Overall, although feeding CPSC to dairy cows did not affect the total SFA or PUFA in the rumen, it was able to modulate the ruminal biohydrogenation of dietary UFA and modify the rumen FA composition.

### 4.2. Rumen Fermentation and Microbial Protein Synthesis

Once the ability of CPSC to modify the biohydrogenation process was proven, this second part was carried out to investigate whether its UFA may impair the ruminal fermentation and microbial protein synthesis, and compromise its practical application in dairy cows feeding. This is of particular relevance due to the potential impact of altered rumen digestion and microbial protein supply on animal performance.

There is a widespread idea that high levels of dietary PUFA can detrimentally affect ruminal fermentation because they are toxic to rumen microbes, resulting in depressed SCFA production [43,44]. The present study observed that higher amounts of PUFA in the diet of CPSC fed cows did not have a great impact on microbial populations in terms of diversity (alpha and beta diversity indices). Moreover, rumen fermentation was not significantly impaired. The total SCFA production was not affected by CPSC, which is in agreement with results observed when supplementing dairy cows with sunflower seed oil [7] or crushed sunflower seeds [45]. Concerning particular SCFA, earlier studies on the inclusion of sunflower lipids as a source of PUFA in dairy cattle diet reported no variation of the rumen SCFA profile [7] or an increase of the molar proportion of propionate at the expense of acetate due to shifts in rumen microbial communities [44]. In the current study, neither the molar proportion of propionate nor the acetate:propionate ratio significantly changed, suggesting that acetate and propionate-producing bacteria, which are considered to be predominant cellulolytic and amylolytic bacteria in the rumen, may have not been influenced by increased levels of linoleic acid [46]. These results disagree with the increased propionate proportion and the decreased acetate detected as a response to dietary inclusion of CPSC in an in vitro study carried out by Benhissi et al. [10]. Although the differences between the in vitro and in vivo results cannot be precluded, the differences in the control diets’ composition, mainly the higher concentrate proportion and the lower UFA contents of the control diet used by Benhissi et al. [10], might partially explain the variable effects of CPSC on the rumen SCFA profile. Regarding butyrate, the reduction caused by CPSC in the molar proportion of this SCFA was consistent with the effect of sunflower seeds [45]. This decrease is often explained by variations in protozoal population and butyrate-producing bacteria, such as *Eubacterium ruminantium* and *Butyrivibrio fibrisolvens*, known to be very sensitive to the toxic effects of elevated amounts of linoleic acid [47,48]. In this study, bacterial community taxonomy was analyzed. Thus, no conclusion about the protozoal population can be inferred. Regarding the known butyrate- producing bacteria, only the enrichment of *Butyrivibrio* genus in the cows fed the control concentrate was observed when the probability was not adjusted (Appendix A), supporting partially this hypothesis. Nevertheless, genus *Coprococcus* were found to be negatively correlated with butyrate (Appendix A) and significantly enriched when CPSC was fed to cows. On the other hand, genus *Stenotrophomonas* were found to be positively correlated with butyrate (Appendix A) and significantly enriched when CTR was fed to the cows, suggesting these two genera would play a role in the differences observed in butyrate concentrations in this study. 

The microbial protein derived from ruminal fermentation is the main component of a metabolizable protein (MP) reaching the small intestine in lactating dairy cows and, qualitatively, its essential aminoacid profile closely matches the one required for milk synthesis [49]. For this reason, the evaluation of the effect of CPSC feeding on the microbial protein supply would be of particular relevance. The results showed a trend for a higher total PD excretion when CPSC was fed to cows, which suggests a possible increase in the flow of purines and, hence microbial proteins into the intestines due to CPSC feeding. In fact, daily microbial nitrogen supply was significantly increased with CPSC. Similar to these results, previous studies reported marked increases of daily total urinary PD excretion and microbial nitrogen supply in response to the enrichment of ruminant diets with linoleic acid-rich fats, including sunflower lipids. This improvement has been attributed to decreased protozoal counts, and reduced predation and competition from protozoa for growth substrates [50,51,52].

## 5. Conclusions

The feeding of CPSC to dairy cows modified the ruminal fermentation and the biohydrogenation process. These changes in ruminal fermentation and ruminal FA were not associated with changes in microbial diversity or the abundance of dominant populations, but rather might be associated with less abundant genera.

## Figures and Tables

**Figure 1 animals-09-00755-f001:**
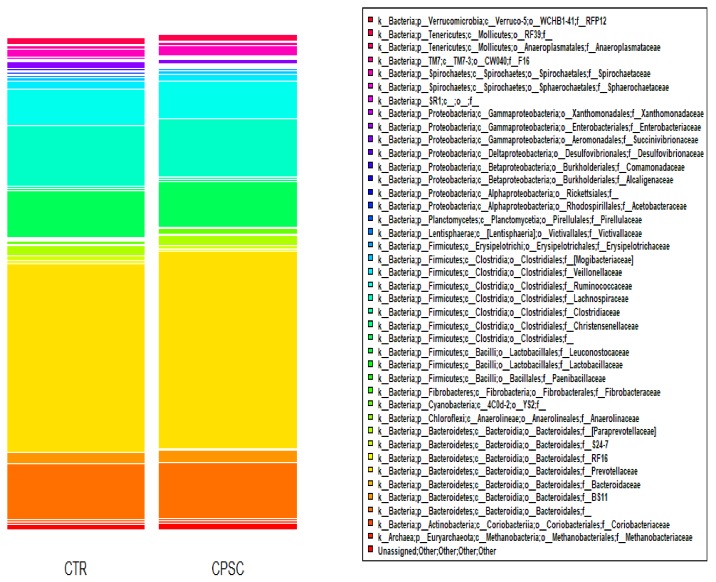
Bacterial community composition at family level in the rumen of the experimental groups: CTR (control), CPSC (cold-pressed sunflower cake).

**Figure 2 animals-09-00755-f002:**
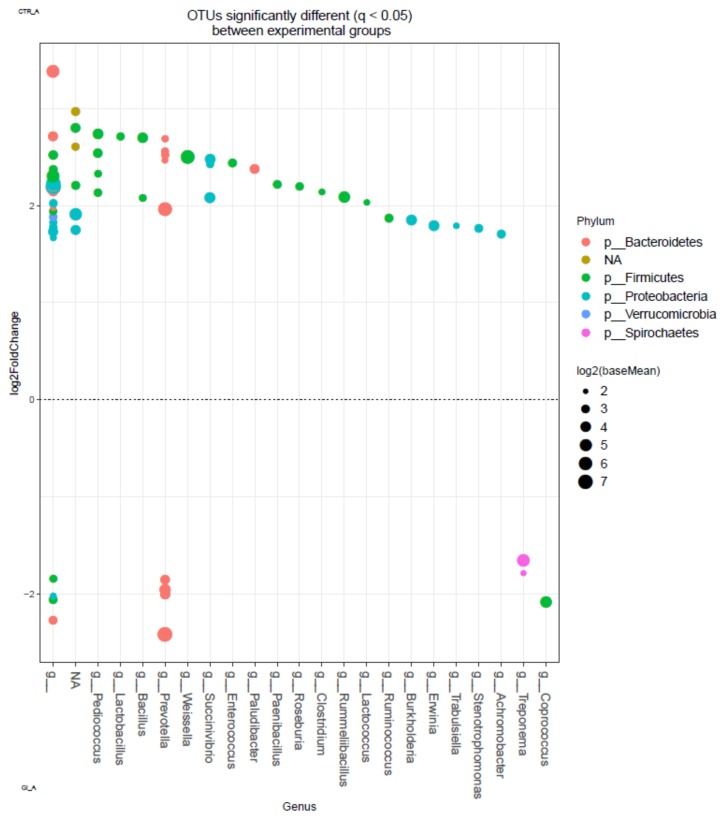
The OTUs at genus level significantly different (q < 0.05) between rumen samples of cows fed control (above) and cold-pressed sunflower cake (below) in the first period of the experiment (n = 5). Each point represents a single OTU colored by phylum and grouped on the x-axis by taxonomy, size of point reflects the log2 mean abundance of the sequence data.

**Figure 3 animals-09-00755-f003:**
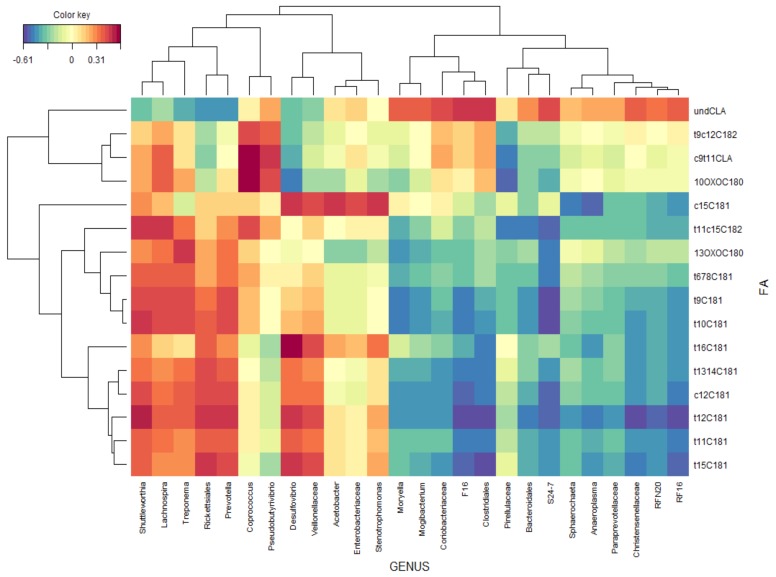
The relationships between clusters of bacterial genus and rumen fatty acids independent of treatment. This clustered image map was based on the regularized canonical correlations between relative bacterial abundances and relative concentrations of rumen fatty acids. Significant correlations are colored following the key shown.

**Table 1 animals-09-00755-t001:** Basal diet, ingredients of concentrates and chemical composition.

*Ingredients (% on DM Basis)*	CTR	CPSC	Basal Diet
Corn	23.7	19.0	
Soybean meal	20.0	15.0	
Cold-pressed sunflower cake	0	23.0	
Palm kernel meal	15.0		
DDGs	14.9	1.0	
Barley	10.8	15.7	
Wheat	6.0	15.0	
Molasses	2.0	2.0	
Hydrogenated palm fat	2.0	0	
Alfalfa pellets	2.0	5.5	
Minerals and vitamins^1^	3.6	3.8	
Maize silage			29
Grass silage			61
Barley straw			9
*Chemical composition*			
UFL	1.02	1.02	
Fat (%)	5.6	5.6	2.8
Acid detergent fibre (%)	9.7	9.8	33.6
Neutral detergent fibre (%)	22.5	19.5	41.1
Acid detergent lignin (%)	2.2	2.3	4.8
Crude protein (%)	19.0	19.0	10.7
Starch (%)	31.7	29.9	5.6
*Fatty acids (% FA)*			
C12:0	7.77	0.12	
C13:0	0.03	0.02	
C14:0	3.09	0.30	
C150	0.04	0.04	
C16:0	23.55	12.05	
C17:0	0.08	0.08	
C18:0	2.90	3.67	
C20:0	0.33	0.34	
C22:0	0.15	0.49	
C23:0	0.06	0.08	
C16:1 cis-9	0.12	0.16	
C18:1 cis-9	25.39	20.56	
C18:1 cis-11	1.13	1.59	
C20:1 cis-11	0.22	0.22	
C18:2n-6	33.04	57.82	
C18:3n-3	1.67	1.53	

CTR: control, CPSC: cold-pressed sunflower cake, FA: fatty acid; ^1^ Contained (g/kg) calcium (270), magnesium (60), sodium (40), phosphorus (40) zinc (5.0), manganese (4.0), copper (1.5); (mg/kg), iodine (500), cobalt (50), selenium (15); (IU/g) retinyl acetate (500), cholecalciferol (100), DL-α-tocopheryl acetate (0.5).

**Table 2 animals-09-00755-t002:** Effect of feeding cold-pressed sunflower cake on ruminal saturated fatty acid composition in dairy cows (n = 10).

Item (g/100 g FA)	CTR	CPSC	SED	*p*-Value
C13:0	0.053	0.054	0.0056	0.875
C13:0 iso	0.056	0.056	0.0065	0.952
C14:0	2.29	1.27	0.083	<0.001
C14:0 iso	0.135	0.152	0.0272	0.558
C15:0	0.793	0.870	0.060	0.241
C15:0 iso	0.245	0.252	0.0391	0.865
C15:0 anteiso	0.695	0.730	0.1078	0.755
C16:0	21.9	15.7	0.53	<0.001
C16:0 iso	0.244	0.303	0.0516	0.286
C17:0	0.434	0.568	0.0174	<0.001
C17:0 iso	0.213	0.220	0.0122	0.585
C17:0 anteiso	0.225	0.237	0.0307	0.704
C18:0	44.0	51.3	0.73	<0.001
C18:0 iso	0.057	0.070	0.0058	0.047
10-oxo-C18:0	0.510	0.571	0.0417	0.180
13-oxo-C18:0	0.233	0.386	0.0208	<0.001
C19:0	0.093	0.106	0.0037	0.007
C20:0	0.750	0.803	0.0136	0.004
C22:0	0.470	0.626	0.0326	0.001
C23:0	0.162	0.181	0.0098	0.084
C24:0	0.528	0.606	0.0135	<0.001
∑ SFA	76.9	76.2	0.83	0.421

CTR: control, CPSC: cold-pressed sunflower cake, SED: standard error of the difference, FA: fatty acid, SFA: saturated fatty acid.

**Table 3 animals-09-00755-t003:** Effect of feeding cold-pressed sunflower cake on ruminal unsaturated fatty acid composition in dairy cows (n = 10).

Item (g/100 g FA)	CTR	CPSC	SED	*p*-Value
C16:1 cis-9	0.068	0.074	0.0085	0.492
C16:1 trans-9	0.012	0.016	0.0024	0.117
C18:1 cis-9	4.25	3.99	0.370	0.512
C18:1 cis-11	0.530	0.580	0.0455	0.297
C18:1 cis-12	0.626	0.995	0.0514	<0.001
C18:1 cis-13	0.130	0.132	0.0134	0.886
C18:1 cis-15	0.216	0.219	0.0099	0.774
C18:1 cis-16	0.123	0.131	0.0057	0.327
C18:1 trans-4	0.197	0.231	0.0154	0.054
C18:1 trans-5	0.129	0.157	0.0123	0.052
C18:1 trans-6-7-8	0.651	0.738	0.0363	0.043
C18:1 trans-9	0.445	0.564	0.0234	<0.001
C18:1 trans-10	0.760	0.944	0.0377	0.001
C18:1 trans-11	4.83	5.56	0.261	0.023
C18:1 trans-12	0.910	1.120	0.020	<0.001
C18:1 trans-13-14	1.18	1.45	0.089	0.016
C18:1 trans-15	0.836	0.963	0.0206	<0.001
C18:1 trans-16	0.761	0.849	0.0433	0.077
C20:1 cis-11	0.153	0.097	0.0199	0.024
C22:1 cis-13	0.039	0.031	0.0058	0.192
C24:1 cis-15	0.098	0.091	0.0063	0.343
C18:2 cis-9 cis-12	2.50	2.39	0.271	0.694
C18:2 cis-9 trans-12	0.053	0.031	0.0045	0.001
C18:2 trans-11 cis-15	0.423	0.627	0.0548	0.006
C18:2 trans-11 trans-15	0.096	0.136	0.0052	<0.001
C18:2 cis-9 trans-11 CLA	0.474	0.429	0.0802	0.589
C18:2 trans-9 cis-11 CLA	0.012	0.009	0.0046	0.536
C18:2 trans-10 cis-12 CLA	0.069	0.066	0.0145	0.858
C18:2 trans-11 trans-13 CLA	0.171	0.227	0.0164	0.009
C18:3n-3	0.642	0.852	0.0965	0.062
C18:3n-6	0.007	0.029	0.0015	<0.001
C20:2n-6	0.026	0.023	0.0041	0.473
C20:3n-6	0.024	0.067	0.0149	0.021
C20:4n-6	0.023	0.056	0.0059	<0.001
∑ MUFA trans	10.70	12.60	0.330	<0.001
∑ MUFA cis	6.20	6.30	0.420	0.794
∑ MUFA	16.90	18.90	0.590	0.009
∑ PUFA	4.77	5.09	0.400	0.446
∑CLA	0.905	0.801	0.0955	0.306
∑ n-3	0.663	0.885	0.0967	0.050
∑ n-6	0.079	0.175	0.0179	<0.001
n-6:n-3	0.117	0.208	0.0243	0.006
C18:1 trans-10:trans-11	0.161	0.172	0.0072	0.165

CTR: control, CPSC: cold-pressed sunflower cake, SED: standard error of the difference, FA: fatty acid, MUFA: mono-unsaturated FA, PUFA: poly-unsaturated FA, CLA: conjugated linoleic acid.

**Table 4 animals-09-00755-t004:** Effect of feeding cold-pressed sunflower cake on ruminal fermentation and urinary purine derivative excretion (LSM, n = 10).

Item	CTR	CPSC	SED	*p*-Value
*SCFA (mmol/ 100 mmol)*				
Acetate	66.5	67.0	0.24	0.108
Propionate	16.9	16.9	0.21	0.940
Isobutyrate	0.90	0.88	0.034	0.696
Butyrate	13.2	12.7	0.11	0.005
Isovalerate	1.06	1.05	0.059	0.835
Valerate	1.41	1.43	0.034	0.546
Acetate:propionate	3.94	3.97	0.061	0.669
*Urinary purine derivatives excretion, mmol/d*				
Allantoin	318	335	8.9	0.093
Uric acid	27	29	2.1	0.439
Creatinine	41	39	2.0	0.434
Total PD	345	364	8.1	0.050
Microbial N flow, g/d	257	273	6.9	0.045

CTR: control, CPSC: cold-pressed sunflower cake, SED: standard error of the difference, SCFA: short chain fatty acid, PD: purine derivatives.

**Table 5 animals-09-00755-t005:** Rumen bacterial community diversity analysis when cows were fed a concentrate with cold-pressed sunflower cake or a control concentrate.

Diversity Indices	CTR	CPSC	SED	*p*-Value
Number of observed OTUs	13432	13703	480.35	0.5901
Good’s coverage	0.953	0.951	0.0027	0.4493
Chao1	22766	23546	468.09	0.1396
Shannon	10.18	10.23	0.193	0.8061

CTR: control, CPSC: cold-pressed sunflower cake, OTU: operational taxonomical unit, SED: standard error of the difference.

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
