# Peer review of "Effect of Feeding Cold-Pressed Sunflower Cake on Ruminal Fermentation, Lipid Metabolism and Bacterial Community in Dairy Cows"

_animals, 2019, doi:10.3390/ani9100755_

Round 1

Reviewer 1 Report

Animals review

General

The intro discusses the potential for altering milk fats by altering rumen fermentation, but many of the cited papers (and others) have shown limited capacity to translate changes in rumen lipids to changes in milk lipids, with the possible exception of slight changes in body lipids linked to all-grass diets.  Can you add anything to convince readers why you expected/didn’t expect to see response?

Unless you can add data on feed intake, milk yield, and milk composition, I think you should limit your discussion to what you actually measured, which was rumen conditions. Along that same line, in the absence of data from urine composition, you have no strong basis to discuss microbial yield or composition. I’m saying, stay within the title of your manuscript.

Specific comments by line

15  “…. flow, all without impairing ….  dominant populations. In the present study, only changes …..”

31 “…. populations; however, they might be … “

114  sterile gauzes

200  Least square means are the same are arithmetic means, unless you have missing data; do you have missing data?

216 Because this is a crossover experiment (all cows were fed both diets), perhaps rewording here and elsewhere to say  “…..when cows were fed either the CTR of the CPSC diets.”  Or just reword, say supplement did not affect ruminal SFA concentrations, and leave the cows out of it.

272 see comment on line 216 – this is confusing, perhaps misleading; it was the same cows fed two treatments in a balanced crossover.  Again, reword to say “….. cows when fed the two dietary treatments.”

299  between treatments are shown in Figure 2  Reword for clarity; you have cows, periods, and treatments, so avoid nebulous terms like ‘experimental groups’.  See title of Figure 3 and Figure 4; I think you mean across treatments in Period 1.

301 ???? significant period effect?  Need explanation of why results from period 1 were selected for presentation.

364 remain uncertain (the subject of the sentence is relationships)

380 data are available

409-422 in the absence of data on milk yield and composition for the cows,  this discussion is not relevant to your results.

453 ??? no urine composition data?  You need to show some numbers, or delete discussion relevant to interpretation of urine composition.

Author Response

Reviewer 1

General

The intro discusses the potential for altering milk fats by altering rumen fermentation, but many of the cited papers (and others) have shown limited capacity to translate changes in rumen lipids to changes in milk lipids, with the possible exception of slight changes in body lipids linked to all-grass diets.  Can you add anything to convince readers why you expected/didn’t expect to see response?

Many authors have mentioned the hypothesis that altering ruminal FA profile would result in a modification of milk FA profile, and not only with grass diets. The intro discusses previous in vitro effects of CPSC on ruminal biogydrogenation (Benhissi eta al., 2016) and previous in vivo effects of CPSC on milk FA profile in sheep (Amores, et al, 2014), and discusses a link between them. We think that reasons that make us expect a response were depicted from L41-51.

Unless you can add data on feed intake, milk yield, and milk composition, I think you should limit your discussion to what you actually measured, which was rumen conditions. Along that same line, in the absence of data from urine composition, you have no strong basis to discuss microbial yield or composition. I’m saying, stay within the title of your manuscript.

Reviewer is right. We have amended the discussion to what we actually measured. We have deleted all references to milk production. Some parameters of urine composition (creatinine) were not in the previous version of the paper. All urine composition data is provided in table 4.

Specific comments by line ü 15  “…. flow, all without impairing ….  dominant populations. In the present study, only changes …..”

The sentence has been rewritten as requested in L15

ü 31 “…. populations; however, they might be … “

The sentence has been rewritten as requested in L31

ü 114  sterile gauzes

The sentence has been rewritten as requested in L114

ü 200  Least square means are the same are arithmetic means, unless you have missing data; do you have missing data?

Yes we have few missing data

ü 216 Because this is a crossover experiment (all cows were fed both diets), perhaps rewording here and elsewhere to say  “…..when cows were fed either the CTR of the CPSC diets.”  Or just reword, say supplement did not affect ruminal SFA concentrations, and leave the cows out of it.

The sentence has been rewritten as requested in L225, and elsewhere.

ü 272 see comment on line 216 – this is confusing, perhaps misleading; it was the same cows fed two treatments in a balanced crossover.  Again, reword to say “….. cows when fed the two dietary treatments.”

The sentence has been rewritten as requested in L282

ü 299  between treatments are shown in Figure 2  Reword for clarity; you have cows, periods, and treatments, so avoid nebulous terms like ‘experimental groups’.  See title of Figure 3 and Figure 4; I think you mean across treatments in Period 1.

The sentence has been rewritten as requested in L304

Figure 3 and 4´s captions have also been changed as requested

ü 301 ???? significant period effect?  Need explanation of why results from period 1 were selected for presentation.

When we analysed the results of the experiment related to bacterial populations, we observed a significant period effect. We realized that because of the design of the trial (crossover), and the long time elapsed between the sampling of ruminal contents of the two periods, the changes in bacterial populations were probably more affected by the changes in the physiological stage of the animals that for the treatment. Because of this only data from period 1 were selected for presentation.

ü 364 remain uncertain (the subject of the sentence is relationships)

The sentence has been rewritten as requested in L364

ü 380 data are available

To our knowledge no data is available about milk fatty acid profile of cows fed cold-pressed sunflower cake (CPSC). There are data about milk fatty acid profile in sheep fed CPSC, and data about effects of sunflower seeds or oil on milk fatty acid profile in cows. We would appreciate very much if the reviewer could provide us with the references we have not be able to find and we will include them in the discussion

ü 409-422 in the absence of data on milk yield and composition for the cows,  this discussion is not relevant to your results.

We have deleted the reference concerning the impact of C18:1 trans10 on human health, but we have not modified the remaining paragraph since it discusses effects observed in the rumen.

ü 453 ??? no urine composition data?  You need to show some numbers, or delete discussion relevant to interpretation of urine composition.

Urine composition data including creatinine concentration is provided in table 4. As you can see, the results are described in the results section in L277-280.

Reviewer 2 Report

Dear authors. This is to my opinion a very valuable manuscript. I have only some minor issues and recommendations for revision.

Overall: Minor English editing is required, especially regarding the use of commas.

Please make sure that all used abbreviations are introduced when first mentioned in the text (e.g., MUFA).

Abstract: Please indicate where you found significances.

L. 108 and elsewhere: "rumen samples" sounds strange. Should better be e.g. "samples of ruminal content".

Section 2.3.2.: Please provide more detail on HPLC analysis, to allow reproduction without consulting further literature.

Section 2.3.3.: It is not clear for me whether you used HPLC or GC. In some parts, it reads to me like the description of GC. Please make this clear. Which detector you used?

Section 2.3.4.: Again, please provide more detail. ... "same FAME mixtures used for the analysis of feeds ..." I cannot find a description of FAME analysis in the feed analysis section (Section 2.3.1.) nor elsewhere.

L. 175: 16S rRNA genes.

Section 2.4.: Which version of SAS did you use? In the model description: "concentrate" should better be "treatment" to prevent confusion; what is "sequence"?; how did you model "cow within pair"?; can you justify such a complex model on the basis of such a low no. of animals?; you probably could just use a fixed treatment effect, random animal effect, and include period and breed as covariables (or remove them) ... just think of it.

L. 229 and elsewhere: Check that units are indicated.

L. 230 and elsewhere: Better use "concentration" instead of "content".

L. 244 and elsewhere: "CPSC-cows" sounds strange. Please use "CPSC supplemented cows" or a similar wording.

Fig. 1: Please provide a better solution, because it is otherwise not readable.

L. 299: "were showed" must read "were shown". Again, please check the whole text for such details.

Fig. 3: Please provide a better solution. And please reflect if you really need this figure. I think it is not necessary.

L. 360-361: Please better explain that sentence.

Author Response

Reviewer 2

Dear authors. This is to my opinion a very valuable manuscript. I have only some minor issues and recommendations for revision.

Overall: Minor English editing is required, especially regarding the use of commas.

ü Please make sure that all used abbreviations are introduced when first mentioned in the text (e.g., MUFA).

We revised the manuscript for abbreviations and only found one abbreviation not introduced (MUFA). This abbreviation has been introduced properly in L237.

ü Abstract: Please indicate where you found significances.

Significances were indicated in the abstract

ü 108 and elsewhere: "rumen samples" sounds strange. Should better be e.g. "samples of ruminal content".

“ruminal samples” has been changed as requested in L108-109

ü Section 2.3.2.: Please provide more detail on HPLC analysis, to allow reproduction without consulting further literature.

This paragraph has been rewritten in order to provide a more detailed description in L137-144.

ü Section 2.3.3.: It is not clear for me whether you used HPLC or GC. In some parts, it reads to me like the description of GC. Please make this clear. Which detector you used?

The Reviewer is correct; there is a mistake in the paper. The analysis was performed with GC equipped with a FID detector. We change the text accordingly in L147-148.

ü Section 2.3.4.: Again, please provide more detail. ... "same FAME mixtures used for the analysis of feeds ..." I cannot find a description of FAME analysis in the feed analysis section (Section 2.3.1.) nor elsewhere.

The Reviewer is correct; we have included a description of FA analysis in feed in the material and methods´ section in L130-135. More detail about FA analysis in samples of rumen content has been also provided in L165-167.

ü 175: 16S rRNA genes.

The sentence has been rewritten as requested in L184

ü Section 2.4.: Which version of SAS did you use? In the model description: "concentrate" should better be "treatment" to prevent confusion; what is "sequence"?; how did you model "cow within pair"?; can you justify such a complex model on the basis of such a low no. of animals?; you probably could just use a fixed treatment effect, random animal effect, and include period and breed as covariables (or remove them) ... just think of it.

SAS version used was Enterprise (2017) and can be found in reference number 29

In the model description “concentrate” was replaced with “treatment” as requested in L208

Sequence is the order the treatments were assigned to each animal across the crossover trial (i.e. CTR_CPSC or CPSC_CTR)

“cow within pair” refers to the nested effect of each animal corresponding to each pair. To make the groups for the trial we grouped animals in pairs according to breed, days in milk, parity and milk yield, obtaining 5 different pairs. Then, each animal of a pair was randomly assigned to one of the two experimental groups.

We think that the proper way to analyse a crossover design is taking into account the effects of period, sequence and treatment, as is described in (cita). If we don´t take into account the period or sequence effects a carry over effect cannot be ruled out.

229 and elsewhere: Check that units are indicated.

Units have been indicated in L239 ,L256, L257.

ü 230 and elsewhere: Better use "concentration" instead of "content".

“Content” has been replaced with “concentration” in L238 and elsewhere.

ü 244 and elsewhere: "CPSC-cows" sounds strange. Please use "CPSC supplemented cows" or a similar wording.

“CPSC-cows” was reworded in L254 and elsewhere

ü 1: Please provide a better solution, because it is otherwise not readable.

A better solution was provided.

ü 299: "were showed" must read "were shown". Again, please check the whole text for such details.

“were showed” was replaced with “were shown” in L304. The whole text was also checked.

ü 3: Please provide a better solution. And please reflect if you really need this figure. I think it is not necessary.

The figure 3 has now been included as supplementary material (Supplementary Figure2)

ü 360-361: Please better explain that sentence.

The sentence was reworded in L358-360

Round 2

Reviewer 1 Report

You have done an acceptable job of addressing my concerns and suggestions, with one exception: the reason for using OUT from Period 1 only, given in your response, should be added to the methods section after  revision, and the section on statistical analysis should describe the model used for those data – it can’t be a crossover with only one period.  I infer from your response to my comments that you don't have data on individual cow feed intake or milk production. If that's not the case, then you should add those data to the revision.